# Fano Resonance Thermo-Optic Modulator Based on Double T-Bus Waveguides-Coupled Micro-Ring Resonator

Hongpeng Li [1,2], Lidan Lu [1,2,]*, Guang Chen [1,2], Shuai Wang [1,2], Jianzhen Ou [3] and Lianqing Zhu [1,2,]*

1 Key Laboratory of the Ministry of Education for Optoelectronic Measurement Technology and Instrument, Beijing Information Science & Technology University, Beijing 100192, China; 2021020161@bistu.edu.cn (H.L.); guangchen@bistu.edu.cn (G.C.); 2021020151@bistu.edu.cn (S.W.)
2 Guangzhou Nansha Intelligent Photonic Sensing Research Institute, Guangzhou 511462, China
3 School of Engineering, RMIT University, Melbourne, VIC 3001, Australia; jianzhen.ou@rmit.edu.au
* Correspondence: lldan_dido@bistu.edu.cn (L.L.); zhulianqing2020@126.com (L.Z.)

**Abstract:** For the silicon optical computing chip, the optical convolution unit based on the micro-ring modulator has been demonstrated to have high integration and large computing density. To further reduce power consumption, a novel, simple Fano resonant thermo-optic modulator is presented with numerical simulation and experimental demonstration. This designed Fano resonator comprises double T-shaped waveguides and a micro-ring with a radius of 10 µm. Compared with the free use of bus waveguides, our double T-shaped waveguides generate a phase shift, along with a Fano-like line shape. The experimental results show that the resonant wavelength shift of the designed modulator is 2.4 nm with a driven power of 20 mW. In addition, the maximum spectral resolution and the extinction ratio are 70.30 dB/nm and 12.69 dB, respectively. For our thermo-optic modulator, the optical intensity power consumption sensitivity of 7.60 dB/mW is three times as large as that of the micro-ring modulator. This work has broad potential to provide a low-power-consumption essential component for large-scale on-chip modulation for optical computing with compatible metal oxygen semiconductor processes.

**Keywords:** thermo-optic modulator; Fano resonance; micro-ring resonator; optical computing

## 1. Introduction

In recent years, with the gradual failure of Moore's law, optical computing has rapidly developed, which can break through the limitations of traditional electronic computers and improve the energy efficiency ratio [1–4]. Shen Y. C. et al. first proposed a fully connected neural network based on the Mach–Zehnder interferometer (MZI) photonic circuit [5]. However, its calculation density is lower than the wavelength division multiplexing (WDM) based on micro-ring resonators (MRRs). Zhao X. et al. successfully demonstrated an efficient MRR temperature sensor on the $TiO_2$ platform. Still, it is unsuitable for large-scale production with Multi-project Wafers (MPWs) [6]. Hsu W.C. et al. demonstrated an MRR-based WDM scheme with high energy efficiency and computation density [7]. As the MRR modulator has the advantages of compact structure, high integration, low insertion loss, and low crosstalk, it has a bright future for application in optical computing.

Fano resonance can cause the optical intensity to abruptly change from 0 to 1 near the resonance wavelength, enhancing optical spectrum resolution (SR). Hu T. et al. proposed the incorporation of a single MRR in conjunction with a cross-waveguide as the coupling point between the upper and lower arms of an MZI. This configuration offers the advantage of achieving a significantly enhanced Fano resonance slope, effectively mitigating the issue of diminished extinction ratio attributed to the dispersion of Fano resonance wavelengths [8]. It is worth noting that the larger size of this structure poses challenges in achieving higher levels of integration. Zhang J. et al. achieved low-power electro-optic modulation using independent silicon nanobeam cavities to generate Fano

resonance [9]. Gu et al. implemented a Fano resonator with a micro-ring and an air hole bus waveguide [10], which exhibits high integration capability, but it is not compatible with the prevailing 180 nm process.

To fabricate the Fano resonator with MPWs, L. Lu et al. have demonstrated by simulation that the micro-ring-coupled double T-bus waveguide has a higher spectral resolution for temperature sensors [11]. Based on all the above research, this paper proposes and tests an efficient thermo-optic modulator composed of double T-bus waveguides coupled with a micro-ring. A Fabry–Perrot cavity as a reflection unit of the double T-waveguides can significantly change the phase [12]. Due to the asymmetric line shape of the Fano resonance, the integrated component has a higher quality factor and a steeper slope compared to the traditional Lorentzian line shape. At the same time, compared with the conventional MRR modulator, our novel Fano resonance thermo-optic modulator has lower power consumption by both numerical simulation and experimental demonstration. Owing to the Fano line shape, our component of T-waveguide-coupled MRRs not only reserves the advantages of the MRR as an optical computing convolution unit but also further reduces power consumption.

This paper is organized as follows. First of all, the design of a double T-waveguide-coupled MRR and the theoretical analysis of the Fano resonance are presented in Section 2. Moreover, a co-simulation for the double T-waveguide-coupled MRR is performed using FDTD and HEAT modules (Ansys Inc., Canonsburg, PA, USA) in Section 3. Then, the optical and electronic microscopic images of the Fano resonator thermo-optic modulator are illustrated in Section 4. The transmission spectra for MRRs within and without double T-waveguide coupling are compared experimentally, and the SR and optical intensity power consumption sensitivity (OIPCS) are calculated based on the acquisition of experimental data. Finally, the future potential applications for this designed component are summarized in Section 6.

## 2. Device Design and Theoretical Analysis

The structure of the designed Fano resonator is shown in Figure 1a, which consists of double T-bus waveguides coupled with a micro-ring resonator (MRR). The cross-section of the MRR is shown in Figure 1b. The Fano resonator device is fabricated on the standard silicon on insulator (SOI) platform. The thickness of the top layer of silicon is 220 nm, and the buried layer of silicon oxide is 2 μm. The thickness of the silicon substrate, the length of the straight waveguide in the raceway micro-ring, the height of the T-waveguide, and the distance between the double T-waveguides are 700 μm, 2 μm, 1.0 μm, and 1.55 μm, respectively. To work in the state of the fundamental mode transmission, the width of the bus waveguide is 450 nm. To obtain low bending loss transmission of the mode, the radius of the raceway micro-ring is set as 10 μm.

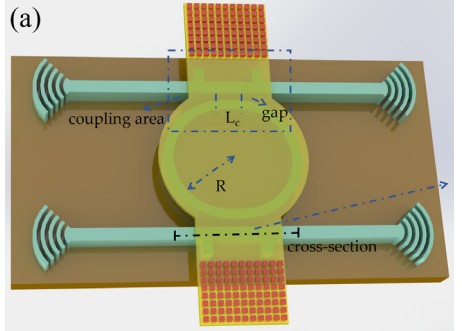
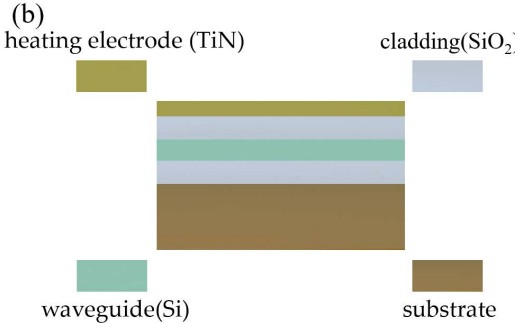

**Figure 1.** (**a**) Schematic diagram of the component structure. (**b**) Cross-section of MRR.

Fano resonance is formed through the coupling between a continuous-mode cavity and a discrete-mode cavity. In the double T-waveguide-coupled micro-ring, the discrete-mode micro-ring and the continuous-mode double T-bus waveguide are coupled to form

Fano resonance. The insertion of double T-waveguides causes a wavelength phase shift, while the discrete resonant mode experiences no additional phase shift. Thus, the phase difference between the discrete and continuous modes is no longer an integer multiple of $2\pi$, resulting in an asymmetrical Fano resonance linearity.

Generally, the coupled-mode theory (CMT) [13] is used to analyze the Fano resonance phenomenon for silicon photonic components. One integrated component structure can be disassembled into basic silicon photonics component parts that all have their own transmission matrix. Furthermore, the output performance of the component can be obtained by the multiplication of their own matrix. In this paper, the Transfer Matrix Method (TMM) is utilized to analyze the double T-shaped waveguide coupled with the MRR structure. When the incident optical field propagates in the bus waveguide with $E_{in} = E_0$, the output electrical field $E_{out}$ in the bus waveguide MRR structure can be expressed by Formula (1) [10]:

$$E_{out} = tE_0 + i\kappa_1 E_1 + i\kappa_2 E_2 + \cdots = \left( t - \frac{\kappa^2 \alpha e^{i\delta}}{1 - t\alpha e^{i\delta}} \right) E_0, \tag{1}$$

where $t$ and $\kappa_1$ ($\kappa_2$) are the coupling region's transmission and coupling coefficients, respectively. Here, $\alpha$ is the linear loss coefficient, and $E_1$ and $E_2$ represent the electrical field from incident light that runs through one circle and two circles in the MRR. The continuous state light is transmitted in the bus waveguide, represented with $tE_0$ in Formula (1). The optical field becomes $E_1 = i\kappa\alpha E_0 e^{i\delta}$ when the light propagates one circle in the micro-ring and returns to the coupling region, where $\delta$ is the round-trip phase shift. $\delta$ can be represented as $\delta = 2\pi n L_R/\lambda$, where $n$ represents the effective refractive index of the waveguide, $\lambda$ is the working wavelength, and $L_R$ is the perimeter of the micro-ring resonator.

When incident light passes through double T-shaped waveguides, the bus waveguide's continuous propagating mode introduces phase $\Delta\Phi$. Thus, $tE_0$ is transferred to $te^{-2i\Delta\Phi}E_0$. Without changing the MRR structure, the optical field in the discrete state does not introduce phase shift. Therefore, the output field of MRR coupled with a double T-shaped waveguide can be expressed as follows:

$$E_{out} = \left( te^{-2i\Delta\Phi} - \frac{\kappa_1 \kappa_2 \alpha e^{i\delta}}{1 - t\alpha e^{i\delta}} \right) E_0, \tag{2}$$

and the wavelength shift of the final transmission spectrum is as follows:

$$T(\lambda) = \left| \frac{E_{out}}{E_{in}} \right|^2 = \left| te^{-2i\Delta\phi} - \frac{k_1 k_2 \alpha e^{i2\pi n L_R/\lambda}}{1 - t\alpha e^{i2\pi n L_R/\lambda}} \right|^2, \tag{3}$$

The device, shown in Figure 1, can be used as a thermal-modulated convolution unit. A thermal resistor (TiN) is used as a thermal source, covered on the T-waveguide-coupled micro-ring, leaving a suitable gap between them. As the voltage gradually increases, the current in the micro-ring modulator also increases accordingly, generating Joule heat in silicon (Si), causing a change in the refractive index of silicon, resulting in a shift in the resonance wavelength. Micro-ring thermo-optic modulation originates from the thermo-optic effect of the silicon material itself. The resonant wavelength change $\Delta\lambda$ caused by temperature change can be expressed as follows:

$$\Delta\lambda = \frac{\lambda \Delta T}{n_g} \left( \frac{\partial n}{\partial T} + \frac{n}{L} \frac{\partial L}{\partial T} \right), \tag{4}$$

where $n_g$ is the group refractive index of the resonator, $L$ is the perimeter of the micro-ring, and $\partial n / \partial T$ is the thermo-optic coefficient of silicon (Si) ($1.86 \times 10^{-4}$/K) [14].

Taking advantage of the high slope of the Fano linear edge, power sensitivity can be improved, and the limitation of the slope ratio of MRRs can be overcome. Moreover, power sensitivity is expressed as follows:

$$S_p = k \cdot S_\lambda, \tag{5}$$

where $k$ is the slope ratio of the steep Fano edge, and $S_p$ is the optical intensity power consumption sensitivity (OIPCS). Additionally, $S_\lambda$ is the wavelength power consumption sensitivity.

## 3. Numerical Simulation

Based on the theoretical analysis above, to obtain the Joule heating effect of the double T-waveguide-coupled micro-ring component, one HEAT module is used to establish a double T-waveguide-coupled raceway micro-ring model to analyze its temperature distribution. The structure of the heating electrode of TiN is shown in Figure 1. The TiN work function is 4.65 eV, and its thermal conductivity is 19.2 W/(m·K). As shown in Figure 2a,b, when an electrical power of 9.12 mW is applied to the TiN electrode, the highest temperature inside the component reaches 363 K, with an average temperature of 314 K.

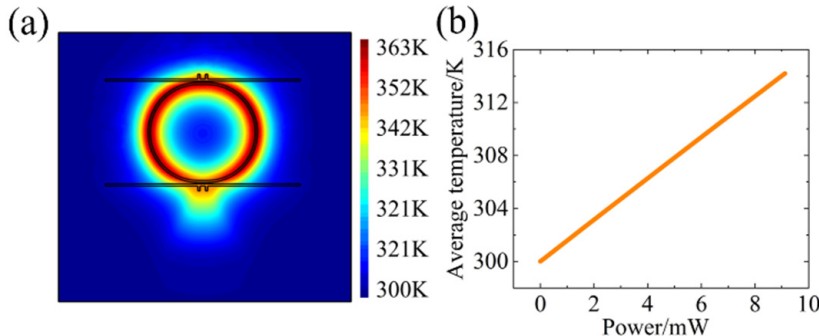

**Figure 2.** (**a**) Temperature distribution of double T-waveguide-coupled micro-ring under 0.7 V. (**b**) The relation between power applied on TiN electrode and device average temperature.

The HEAT module is used to simulate the relation between the voltage and temperature of the device and then imported into the FDTD module. The Fano resonance normalized transmission spectrum of the double T-micro-ring with a wavelength from 1510 nm to 1600 nm with a change in voltages from 0 to 0.7 V is shown in Figure 3a, which indicates an obvious Fano resonance with the change in driven voltages. As the voltage increases, there are gradual redshifts for the Fano resonance spectrum. The result shows that the wavelength of the double T-waveguide-coupled micro-ring has shifted by 4.45 nm at a voltage of 0.7 V, as shown in Figure 3b. The electrical field distribution of the optical signal at a resonant wavelength of 1581.38 nm and a non-resonant wavelength of 1587.35 nm for the double T-waveguide-coupled micro-ring are shown in Figure 3c,d, respectively.

According to the simulation results from the HEAT module in Figure 2, it can be observed that the square of the driven voltage and the average temperature for the double T-waveguide-coupled micro-ring have a linear relationship, and the relation between the silicon refractive index and temperature is linear. Therefore, the relation between the square of the voltage and wavelength redshift is also linear. Owing to the thermo-optic effect-induced change in silicon refractive index, the temperature variation of the component causes significant wavelength shifts of the resonance peak for the micro-ring.

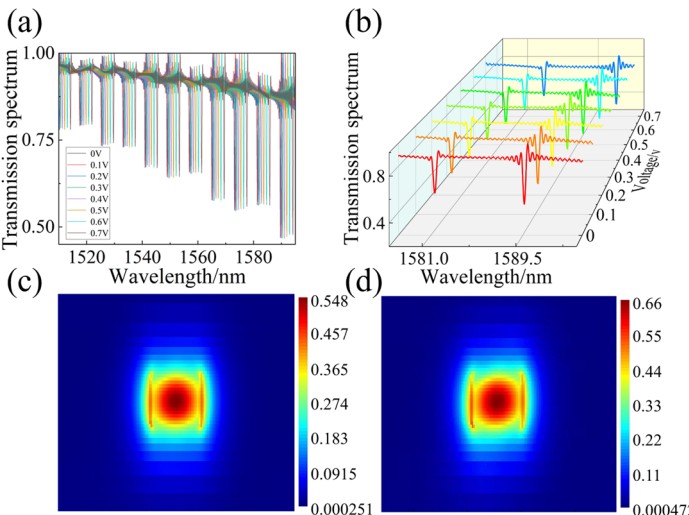

**Figure 3.** (**a**) Normalized transmission spectra of double T-waveguide-coupled micro-ring as voltages changing from 0 to 0.7 V. (**b**) Normalized transmission spectrum from 1580 nm to 1588 nm for the double T-waveguide-coupled micro-ring. (**c**) The electrical field of optical signal at a resonant wavelength of 1581.38 nm. (**d**) The electrical field of optical signal at a non-resonant wavelength of 1587.35 nm.

## 4. Experimental Demonstration

The double T-waveguide-coupled micro-ring is fabricated based on the above design, and images taken from the optical and scanning electronic microscopes are shown in Figure 4a,b, respectively. Although some fabrication errors may occur during the process, such as large roughness and comparatively high loss, they can be ignored while only considering the spectrum shift with the change in applied voltage.

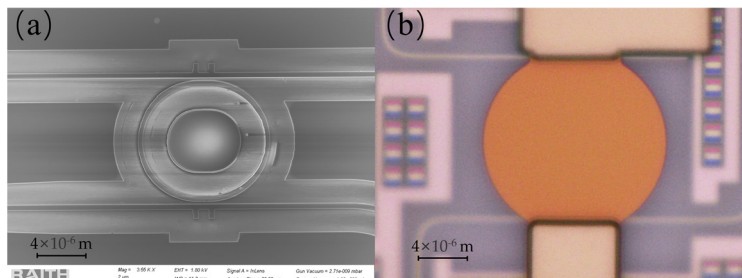

**Figure 4.** (**a**) Scanning electronic microscope image of double T-waveguide-coupled raceway micro-ring without heating electrode. (**b**) Optical microscope image of double T-waveguide-coupled raceway micro-ring with heating electrode.

To compare the performance between the basic MRR and components with T-waveguide in the experiment, the experimental verification is carried out for MRRs with the help of a semi-automatic coupling platform and probe. As shown in Figure 5a, a normalized transmission spectrum ranging from 1530 nm to 1600 nm is measured by an optical spectral analyzer (OSA). As shown in Figure 5b, voltages from 0 to 0.75 V are applied to the TiN electrode, and the normalized transmission spectra are obtained from 1581 nm to 1583 nm, which excludes the spectra of the light source, the optical link, and the grating coupler. As shown in Figure 5c, the relation between driven voltage and current is measured for the untuned micro-ring, and the total resistance between the two electrodes is calculated from the voltage–current curve, which is about 41.66 Ω. Based on the above spectra and experimental data, the relation between power and wavelength is calculated as illustrated in Figure 5d, and a similarly linear relation between power and wavelength is obtained.

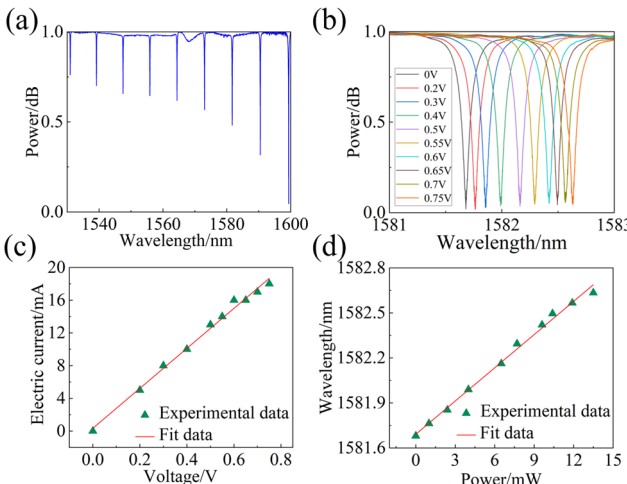

**Figure 5.** (**a**) Full normalized transmission spectrum for micro-ring without T-waveguide coupling (**b**) Normalized transmission spectrum at driven voltage from 0 to 0.75 V for micro-ring without T-waveguide coupling. (**c**) The relation between driven voltage and current for the same component. (**d**) Relation between wavelength shift of resonant peak and driven power for the above component.

A semi-automatic temperature-controlled coupling platform where the voltages are applied by the probes to the double T-waveguide-coupled micro-ring is used. Except for adding the double T-waveguide, the structure of the component is the same as that of normal MRRs. As shown in Figure 6a, the full normalized transmission spectrum ranging from 1530 nm to 1600 nm is measured by OSA. As shown in Figure 6b, the normalized transmission spectra from 1581 nm to 1585 nm are selected by applying voltages ranging from 0 V to 0.75 V, which excludes the spectra of the light source, the optical link, and the grating coupler.

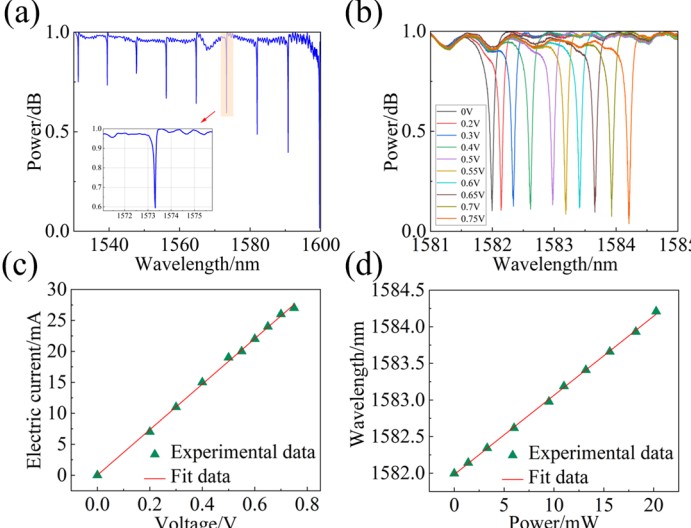

**Figure 6.** (**a**) Full normalized spectrum of double T-waveguide-coupled micro-ring. (**b**) Normalized transmission spectra with a voltage changing from 0 to 0.75 V. (**c**) Relation between driven voltage and current for the same component. (**d**) Relation between wavelength shift of resonant peak and driven power for the above component.

Compared with the normal MRR, the double T-waveguide-coupled micro-ring has higher wavelength power sensitivity, along with lower power consumption for multiplication operations in convolution. The relation between voltage and current for the double T-waveguide-coupled micro-ring is also obtained in Figure 6c, which shows that there is a similarly linear relationship between current and applied voltage. This experimental result

is consistent with previous numerical simulation results. As shown in Figure 6c, the total resistance between the two electrodes is calculated, which is about 36.75 Ω. Moreover, it is shown in Figure 6d that the power–wavelength relation is calculated from the above experimental results and data, revealing a similarly linear relationship between power and wavelength.

## 5. Discussion

The silicon effective refractive index can be changed by the variation in temperature due to the thermo-optic effect. Even a slight variation in temperature can change the resonant peak of the transmission spectrum for MRRs. For applications in optical computing, it is necessary to obtain the relation between power consumption and light intensity. Therefore, the formula of OIPCS is derived as follows:

$$\frac{d\Delta P_o}{d\Delta P_e} = \frac{d\Delta P_o}{d\Delta \lambda} \cdot \frac{d\Delta \lambda}{d\Delta P_e} = SR \cdot \frac{d\Delta \lambda}{d\Delta P_e}, \tag{6}$$

where $d\Delta P_o$, $d\Delta P_e$, and $d\Delta \lambda$ are optical intensity, power consumption, and wavelength, respectively. Here, $dP_o/dP_e$ represents the OIPCS, which is obtained by the multiplication of SR and wavelength power consumption sensitivity (WPCS) for Fano resonators. According to Formula (6), the experimental results of the MRR and the double T-waveguide-coupled micro-ring and the SR of the normal MRR and the double T-waveguide-coupled micro-ring are 35.20 dB/nm and 70.30 dB/nm, respectively. The OIPCS of the normal MRR is 2.60 dB/mW, and the WPCS is 0.074 nm/mw. However, the OIPCS of the double T-waveguide-coupled micro-ring is 7.60 dB/mW, and the WPCS is 0.108 nm/mW. Obviously, the SR, OIPCS, and WPCS of the double T-waveguide-coupled micro-ring are significantly improved compared with those of the normal MRR. The reason why there are obvious performance improvements for the double T-waveguides component is mainly that the optical intensity of the Fano resonator changes sharply from 0 to 1 near the resonant peak of the transmission wavelength. For the same change in optical intensity, the power consumption of the double T-waveguide-coupled micro-ring requires only 1/3 as much as that of normal MRRs for the resonant wavelength, which brings higher modulation efficiency and is more suitable for convolution units in optical computing.

## 6. Conclusions

In this paper, a novel essential component based on a micro-ring resonator for silicon optical computing convolution units is proposed. By adding double T-bus waveguides, a thermal–optical modulator is obtained with a Fano resonance linear transmission spectrum. A large phase shift is obtained by comparatively low driven power around 20 mW in the TiN electrode, where voltage changes generate Joule heat, resulting in a change in the silicon refractive index as well as a wavelength shift for the Fano resonance. Numerical simulation and experimental results demonstrate that the Fano resonance line-shaped transmission spectrum for the double T-waveguide-coupled micro-ring has higher spectral resolution (SR) and optical intensity power consumption sensitivity (OIPCS) than those of the MRR transmission spectrum, which significantly reduces the modulation power required. Besides optical computing, this work also has tremendous potential fields for applications, such as optical switching, optical sensing, and optical communications. In the future, the quality factor of T-shape waveguides can be further improved to enhance the SR, along with superior component performance.

**Author Contributions:** Conceptualization, H.L. and L.L.; methodology, H.L. and L.L.; software, H.L. and S.W.; validation, H.L. and G.C.; formal analysis, H.L. and J.O.; writing—original draft preparation, H.L.; writing—review and editing, H.L.; supervision, L.L. and L.Z. All authors have read and agreed to the published version of the manuscript.

**Funding:** This work is supported by National Natural Science Foundation of China (62205029) and Young Elite Scientists Sponsorship Program by CAST (2022QNRC001).

**Institutional Review Board Statement:** Not applicable.

**Informed Consent Statement:** Not applicable.

**Data Availability Statement:** Data are contained within the article.

**Conflicts of Interest:** The authors declare no conflicts of interest.

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
