# Peer review of "Fano Resonance Thermo-Optic Modulator Based on Double T-Bus Waveguides-Coupled Micro-Ring Resonator"

_photonics, doi:10.3390/photonics11030255_

Round 1

Reviewer 1 Report

Comments and Suggestions for Authors

This paper proposes and discusses a thermooptic modulator based on a double T-bus waveguide coupled with a micro-ring resonator. The paper has the potential to be published in Photonics. My comments are as follows:

1) The keyword “T-bus waveguides coupled with a micro-ring resonator” should be in the title of the paper.

2) It would be helpful for readers if the principle of Fano resonance in the proposed T-bus waveguides coupled with an MRR could be concisely explained.

3) What advantages does the proposed modulator have compared with other Fano resonance MRR devices (for example, [10])?

4) The authors mention, “the width of the bus waveguide is 450 nm,”. What is the width of the MRR waveguide? Is it the same as that of the bus waveguide?

5) In Figs. 3, 5, and 6, the voltages are shown as a variable parameter, but the results should be discussed not regarding a “voltage” but regarding “(injected) power”.

6) The explanations about OIPCS are repeated on pages 3 and 6. Does Sp in (5) mean dPo/dPe in (6)? The parameters should be unified in both (5) and (6).

7) What are the perimeter of the MRR, coupling coefficient, and loss coefficient of the MRR used in the simulation and experiments?

8) What is the reason that the resistances of both devices are different? Both devices should have the same heater structure.

9) Why is WPICS of the normal MRR different from that of the double T-waveguide with the MRR? The wavelength shift should only depend on the perimeter of the Si MRR.

Minor revisions

10) Kappa 1 and kappa 2 are missing in (1).

11) What does “TTM” on page 3 stand for?

12)“ the output electrical Eout” -> “the output electrical field Eout”

13) “dP0” in (6) is not optical intensity but the change in the optical intensity. dPe, dP0, d lambda also have the same issues.

Comments on the Quality of English Language

I recommend that you have a professional English proofreading.

Reviewer 2 Report

Comments and Suggestions for Authors

The paper presents a novel approach to enhancing the efficiency of silicon optical computing chips by introducing a Fano resonant thermo-optic modulator, composed of double T-shaped waveguides and a micro-ring resonator (MRR). The paper utilizes coupled-mode theory (CMT) to analyze the Fano resonance, providing a comprehensive theoretical foundation. The experimental demonstration further validates the efficacy of the proposed modulator. The results exhibit a significant reduction in power consumption, with a clear enhancement in spectral resolution and extinction ratio. The paper effectively introduces the status of the field and presents its findings with remarkable clarity. I have only a minor question about the wavelength range studied in the paper. The Fano resonance is designed to be at around 1500 nm for the purpose of telecommunication. For other applications that operate at visible range, can this design be extended to other wavelengths?

Reviewer 3 Report

Comments and Suggestions for Authors

The authors demonstrated a waveguide - microring resonator coupled system, in which the waveguide has a large T-shaped cross-section while the microring features a standard cross-section as a silicon wire. This asymmetric waveguide design results in a fano resonance and offers one sharp edge in the spectrum for efficient thermo-optic tuning. The article is organized in a logical way and the results match well with the expectations. Nevertheless, waveguide-ring coupled filters / switches have been extensively studied in the last two decades. The findings in this work do not deliver a significant advancement to this field and should be major revised to appeal to the specialists readers.

1. The waveguide structure should be clearly displayed in Fig. 1, with detailed sketches of the cross-sections at different parts as separate figures.

2. Is the T-shaped waveguide multimode? How to suppress higher-order modes from coupling with the ring in practice? 

3. The tuning efficiency depends on the thermo-optic coefficient, the quality factor / spectral width of the coupled system and the extinction ratio (critical coupling) between on and off states. The authors should make this theoretical analysis clear and show how a fano resonance is potentially superior to a high-Q ring and other methods in creating a sharp drop edge, e.g., using coupled rings.

Comments on the Quality of English Language

The English language is OK.

Reviewer 4 Report

Comments and Suggestions for Authors

In this manuscript, the authors propose a thermo-optic modulator comprising a micro-ring resonator and T-shaped waveguides. They claim that the proposed structure has lower power consumption compared to a directly coupled ring resonator. However, the figures presented are confusing, and the theoretical analysis appears incomplete. Therefore, I would not recommend publication in its current form. Major modifications may be necessary for reconsideration. Detailed technical comments are provided as follows.

1. In Figure 1, it looks like the structure is another material (cladding) when covered by the TiN. I initially thought the T-waveguide is made of the cladding material (the void region) when saw Figure 1.

2. The theoretical model is not complete. I did not see the importance of adding the T-waveguide for the design. Since all the points of the manuscript is to add the T-waveguide, please explain the physics and dimensionality of T waveguide in more detailed ways.

3. Why Figure 5(c)(d) and Figure 6(c)(d) are identical? If the authors would like to compare the performance with and without T-waveguide coupled microring, it might be better to overlay the results in the same plot for comparison.

4. Line 89, TTM abbreviation is not defined when used.

Reviewer 5 Report

Comments and Suggestions for Authors

Recommendation: Publish in Minor, Required Changes

Comments:

In this study, a novel Fano resonant thermo-optic modulator with a double T-shaped waveguide and a micro-ring was proposed to achieve reduced power consumption in silicon photonics-based optoelectronic computing chips. Experimental results demonstrated a shift in the modulator's resonant wavelength by 2.4 nm under an electric power of 20mW, a maximum spectral resolution of 70.30 dB/nm, an extinction ratio (ER) of 12.69 dB, and an optical intensity-power consumption sensitivity of 7.60 dB/mW, surpassing the performance of a traditional microring resonator modulator. These findings hold promise for the development of highly integrated and efficient on-chip modulators.This article is clear, concise, and suitable for the scope of the journal. This article has a few problems, which can be published with minor repairs.

Questions are as follows:

1.      The first occurrence of “TTM” needs to be expanded to its full name.

2.      Equation 3 is missing between Equation 1 and 2.

Round 2

Reviewer 1 Report

Comments and Suggestions for Authors

The manuscript has been revised well. I recommend this paper for publication.

Author Response

Dear reviewer,thank you so much for your revierwing!We deeply appreciate your recognition of our research work.

Reviewer 3 Report

Comments and Suggestions for Authors

Unfortunately, the authors have not clearly addressed my previous comment (3). Except for some hand-waving text, detailed comparison of key performance parameters to existing thermo-optic modulators using symmetric ring-resonators is missing. This work brings too little advancement to the community and should not be published in this form.

Comments on the Quality of English Language

The English language is fine.

Reviewer 4 Report

Comments and Suggestions for Authors

The authors have addressed my concerns. I can recommend for publication.

Author Response

(The authors gave the same response as above.)

Round 3

Reviewer 3 Report

Comments and Suggestions for Authors

The authors have addressed some of my concern. Though the performance in terms of OIPCS is not the state of the art (see Optics Express Vol. 24, Issue 18, pp. 20187-20195 (2016)), and the comparison to other methods of sharpening the spectral response is still missing, the T-shaped waveguide design has been well verified in the resonator-coupled system and could be interesting to colleagues in the field.

Comments on the Quality of English Language

English language is fine.